# Research on Focal Length Measurement Scheme of Self-Collimating Optical Instrument Based on Double Grating

**DOI:** 10.3390/s20092718

**Published:** 2020-05-10

**Authors:** Wenchang Yang, Zhiqian Wang, Chengwu Shen, Yusheng Liu, Shaojin Liu, Qinwen Li, Wen Du, Zhuoda Song

**Affiliations:** 1Changchun Institute of Optics, Fine Mechanics and Physics, Chinese Academy of Sciences, Changchun 130033, China; wenchang201506@126.com (W.Y.); shenchengwu@ciomp.ac.cn (C.S.); liuys@ciomp.ac.cn (Y.L.); liusj@ciomp.ac.cn (S.L.); liqinwen16@mails.ucas.ac.cn (Q.L.); duwen17@mails.ucas.ac.cn (W.D.); songzhuoda17@mails.ucas.ac.cn (Z.S.); 2University of Chinese Academy of Sciences, Beijing 100049, China

**Keywords:** focal length measurement, moiré fringe, self-collimating optical system, mathematical model, double grating

## Abstract

In this paper, we propose a scheme for measuring the focal length of a collimating optical instrument. First, a mathematical model for measuring the focal length of a collimator with double gratings is derived based on the moiré fringe formula and the principles of geometric optics. Mathematical simulation shows that a slight difference in the focal length of two collimators has an important influence on the imaging law of moiré fringes. Our solution has a good resolution ability for focal length differences within 5‰, especially in the small angle range below 4°. Thus, the focal length of collimators can be measured by the amplification of the slight difference. Further, owing to the relative reference measurement, the measurement resolution at the symmetrical position of focal length is poor. Then, in the designed experiment, a corresponding moiré image at different angles is acquired using collimators with known focal length. The experimental results indicate that the root mean square error (RMSE) of the collimator corresponding to grating angles of 2°–4° is better than 4.7‰, indicating an ideal measurement accuracy of the proposed scheme. This work demonstrates that our proposed scheme can achieve an ideal accuracy in the measurement of a symmetrical optical path.

## 1. Introduction

The grating measurement system commonly uses the grating pitch as the reference. When the grating pitch is below 10 μm, it is a grating precision measurement system that follows the principles of diffraction interference of light. When the grating pitch is in the range from 10 to 200 μm, the grating line distance is much larger than the source wavelength and the diffraction phenomenon can be neglected. Thus, the grating measurement can be analyzed by the occlusion effect of geometric optics [1]. At present, grating measurement technologies based on the principles of geometric optics, such as grating rulers, encoders, and roll angle measurement, have been widely used [2,3].

Moiré fringes are generated when two gratings are rotated relative to each other, and the displacement measurement can be realized by counting and subdividing the moiré fringes. The common collimating optical system is often referred to as a collimator, which can simulate the beam from infinity. The collimator is an important tool for optical instrument installation and detection, as well as an indispensable component of optical instruments such as theodolite [4,5,6]. Currently, the collimator is developing towards the directions of lightweight and miniaturization. With regard to self-collimating optical instruments designed for various specialized fields, the focal length is one of the most important parameters of the collimator, and its accuracy plays a significant role in the use of precise optical systems [7,8]. However, the measurement accuracy of long focal length optical systems is easily affected by external factors, and the processing error of the lens in the manufacturing process also has significant influences on the measurement accuracy [1,9,10].

On the basis of previous research theories, current focal length measurement methods are generally divided into four categories. The first category refers to the conjugate method. In this method, a reticle or a Porro tester is placed on a focal plane of a collimator, a digital image formed by the charge coupled device (CCD) is collected through the optical system, and then the focal length can be calculated using the proportional relationship between object and image. This scheme belongs to the conventional method of focal length measurement, and its measurement accuracy is very poor because of the linear object–image relationship. The second category is the laser differential confocal method proposed by Zhao et al. [8,11]. This method grasps the characteristics of the absolute zero position in the axial light intensity response curve and the focus position of the confocal system, and can realize the accurate location of the cat’s eye position and the confocal position. By this method, the measurement resolution and anti-interference ability can be greatly improved, especially for ultra-long focal length measurement. Nevertheless, it has strict requirements on ambient temperature, air disturbance, and vibration, thus limiting its wide applications. The third category of method is based on Fizeau interferometer. It requires a variety of standard emission spheres. The longer the focal length, the more difficult it is to process and detect such standard mirrors [12,13]. The fourth category involves a long focal length measurement method proposed by Kafri based on the Talbot effect and interference technique [14]. The grating technology has received extensive attention from many scholars in the field of measurement owing to its anti-interference and relatively low cost [15,16,17,18]. In recent years, many methods based on the grating interference technology have been proposed one after another [19,20,21,22].

In this study, we propose an effective focal length measurement scheme for collimating optical systems. The main contributions are as follows:(1)Different from previous interference-based measurement schemes in which the period of the diffraction grating is less than 10 μm, our scheme sets the grating pitch as 70 μm, which is much larger than the wavelength of the light source. Hence, the mathematical model of this scheme is based on the principles of geometric optics. We deeply analyze the relationship between the stripe width and the rotation angle of two gratings’ bases on the classical theory of moiré fringes. By introducing the scaling effect of the focal length difference on the grating image, the focal length measurement formula of the optical system is deduced. Theoretical and simulation analysis shows that the focal length difference between the collimators can be effectively amplified by combination of the collimated optical path of the collimator and the moiré fringes generated by two gratings.(2)We carefully design the structure of the entire measurement sensor, and provide the specific operation process in detail. The results show that our solution can effectively improve the focal length measurement accuracy of the collimating optical system.(3)Our scheme is different from the focal length method proposed by Kafri and other scholars. The rotation value of the grating can be acquired by the shaft angle encoder, which effectively eliminates the slope error caused by the grating stripes. Hence, our scheme is more suitable for the batch detection of collimator focal length.

The rest of this paper is organized as follows. In Section 2, we present the relationship model between moiré fringes and the focal length of the symmetric collimator system, and simulate the characteristics of the measurement model. Then, Section 3 introduces the structural design of the measuring setup and the parameters of main components. In addition, the measurement results are analyzed and compared with the existing methods. Finally, the conclusions are drawn in Section 4.

## 2. Methods

### 2.1. Moiré Fringe Equation

The principle model of measuring the focal length of a collimator with moiré fringe is shown in Figure 1. The entire setup consists of two parts, that is, the emitting part and the receiving part. The light source and the G_2_ are fixedly connected to the *z*-axis, and when the control mechanism rotates, G_2_ is driven to rotate synchronously around the *z*-axis. After the whole mechanism is calibrated, the angle of G_2_ rotation can be known by the shaft encoder. The focal length of collimator 1 is defined as the measuring standard, and that of collimator 2 is unknown. The light forms a collimated beam through collimator 2 and passes through the G_1_ to form moiré fringes. The stripe width varies nonlinearly with the angle of rotation, and the final moiré fringe images for different widths are acquired by the CCD.

The focal length and moiré fringe relationship of a symmetric optical system can be derived by the moiré fringe equation. The Cartesian space coordinate system is established as illustrated in Figure 2. It is assumed that there are two gratings, G_1_ and G_2_, and that the grating pitches are *d*_1_ and *d*_2_, respectively. Any grating line of G_1_ is taken as the *y*-axis, and its vertical direction as the *x*-axis. Then, *θ* represents the rotation angle between two gratings, and *W* represents the distance between two stripes. The moiré fringe equation after grating overlap can be obtained by this coordinate system [23,24,25,26,27].

According to the plane geometry principle, the width of moiré fringe can be given by (further details in Appendix A)
(1)$$W=\frac{d_{1}d_{2}}{\sqrt{d_{1}^{2}{+d}_{2}^{2}-{2d}_{1}d_{2}\cos\theta }}$$

In practical engineering, two gratings with the same pitch *d* are usually used [2,26]. Then, the above formula can be simplified as
(2)$$W=\frac{d}{2\sin(\frac{\theta }{2})}$$

Equation (2) indicates that the width *W* of the stripes is nonlinearly related to the angle *θ*. At a small angle *θ*, moiré fringes have a significant amplification effect on the grating pitch *d*.

### 2.2. Collimator Focal Length Equation and Simulation

Assume that the focal lengths of the two collimators are *f*_1_ and *f*_2_, respectively. According to the geometrical optics principles, the scaling effect *σ_d_* of the difference between the two focal lengths on the grating pitch can be expressed as
(3)$$\frac{\sigma_{d}}{d}=\frac{f_{2}-f_{1}}{f_{1}}$$

Owing to the influence of the scaling factor *σ_d_*, the grating pitch is changed from $d_{1}=d_{2}=d$ to the following form:(4)$$\{\;\begin{array}{c} d_{1}=d\\ d_{2}=(\sigma_{d}+1)d \end{array}$$

Substituting Equation (4) into Equation (1) yields the relationship expression between the moiré fringe width and the focal length factor:(5)$$W^{\prime }=\frac{(\sigma_{d}+1)d^{2}}{\sqrt{d^{2}+{\left(\sigma_{d}+1\right)}^{2}d^{2}-2(\sigma_{d}+1)d^{2}\cos\theta }}=\frac{f_{2}d}{\sqrt{f_{1}^{2}+f_{2}^{2}-{2f}_{1}f_{2}\cos\theta }}$$

According to the root of the quadratic equation formula, the focal length *f*_2_ of collimator 2 can be expressed as
(6)$$f_{2}=f_{1}\left(\frac{-W^{\prime 2}\cos\theta \pm W^{\prime }\sqrt{d^{2}-W^{\prime 2}\sin^{2}\theta }}{d^{2}-W^{\prime 2}}\right)$$
where *θ*, *f*_1_, and *d* are known quantities and *W*′ is the fringe width calculated using the acquired image. Then, the focal length *f*_2_ of collimator 2 to be measured can be obtained by substituting *θ*, *f*_1_, *d*, and *W*′ into (6). Equation (6) also shows that there are two solutions for *f*_2_. As the variation of moiré fringe width has a nonlinear relationship with the focal length, the measurement accuracy can be effectively improved using this amplification relationship.

In order to further verify the effectiveness of this scheme, the width variation of moiré fringes caused by the focal length is simulated. Equation (2) corresponds to the situation when the focal lengths of two collimators are equal, and (5) is the change of stripe width when the focal length factor is introduced. By comparing (2) and (5), the corresponding fringe width variation Δ*W* at different grating angles can be expressed as follows:(7)$$\Delta W=W-W^{\prime }$$

At a grating pitch of 70 μm, when the focal length difference of two collimators is 1‰, 3‰, and 5‰, the variation amount Δ*W* of moiré fringe width in the range of 0°–10° with different angles *θ* can be calculated using (7), as shown in Figure 3. When there is a slight difference in the focal length of two collimators, the width of moiré fringe changes significantly. As the grating angle decreases, the variation amount of fringe width increases, and the trend is particularly noticeable at a small angle in the range of 0°–4°. With regard to collimator 1 with a known focal length, as the focal length difference increases, the width variation of moiré fringe also increases rapidly. Therefore, the simulation results show that our scheme still has a good resolution ability when the focal length difference is below 5‰.

In addition, it can be seen from (6) that there are two solutions for collimator 2 to be tested. Hence, it is necessary to verify the measurement resolution in both the positive and negative directions. Figure 4 shows the variation of moiré fringe width corresponding to grating angles of 1° and 2°, respectively. Assume that the difference in the focal length between two collimators is ±5%, ±3%, and ±1%, respectively. Then, the corresponding ratios *f*_2_/*f*_1_ are 0.95 and 1.05, 0.97 and 1.03, and 0.99 and 1.01, respectively, where 0.95, 0.97, and 0.99 mean that the value *f*_2_ of collimator 2 to be measured is smaller than the reference focal length value *f*_1_, and 1.01, 1.03, and 1.05 mean that *f*_2_ is larger than *f*_1_. From the variation of moiré fringe width at different angles, it can be seen that moiré fringe has a small difference in the symmetrical position of the reference focal length, indicating that this method has a poor discriminating ability at the symmetrical position of the reference collimator.

## 3. Experiments and Results

### 3.1. Experimental Setup

Figure 5 shows the layout scheme of the designed focal length measurement experimental setup. The whole setup is placed on the optical platform, and the focal lengths of the two collimators are known. The axes of the two collimators are calibrated to the same axis by the theodolite, and the angle error after calibration is less than 0.5 arcsec. The structural principle of the receiving part is shown in Figure 1. G_2_ is placed on the focal plane of collimator 2. When a monochromatic plane wave is incident upon the periodic Ronchi grating (G_2_), the light intensity behind the grating is repeated at regular distances, and this phenomenon is called the Talbot effect. The regular distance is referred to as the Talbot length *D* = 2m*d*^2^/*λ*, m = 1, 2, 3…, where *d* is the period of grating and *λ* is the wavelength [28,29,30]. In this scheme, G_1_ is placed at the Talbot distance, and the Talbot image and G_1_ generate the moiré fringes that are focused on the CCD by the two collimators. The final image can be processed by a computer.

The main parameters of the experimental setup are shown in Table 1.

Figure 6 presents the structural principles of the emitting part. The entire structure of the grating G_1_, the light source, the rotating mechanism, and the shaft angle encoder are fixedly connected together, which are coaxially connected with the collimator. The CCD used in the experiment has the characteristics of low illumination and high sensitivity, and it can be saturated even if the light source is very weak. When a 5 mW red LED is used as the light source, a clearer moiré fringe can be obtained. The collimator is a common standard optical instrument in the measurement field, and its internal structure is not discussed here. 

The operation process of the experimental setup is as follows. Firstly, the control mechanism is rotated to drive G_2_ and the shaft encoder to rotate, and the generated roll angles can be obtained by the shaft encoder. Then, the width of moiré fringes superimposed on the CCD changes at different angles, and the corresponding images are acquired at 2°, 3°, and 4°, respectively, from the zero position. The rotation angle of the encoder is taken as the true value, and the above operation is repeated 10 times to calculate the corresponding focal length. To reduce the effects of random measurement errors, the root mean square error (RMSE) of the statistical results is calculated here. Finally, the measurement results are compared with real values of collimator 1 to acquire the errors at different angles. Different from the grating focal length measurement method proposed by previous scholars in structural design, our designed setup can acquire the rotation value of the grating by the shaft angle encoder, which effectively eliminates the angle error caused by the calculation of the slope of grating stripes.

The width *W*′ in (6) needs to be calculated from the moiré fringe image collected by this setup. Figure 7a presents a captured moiré image through an image processing series, that is, filtering, contrast enhancement, threshold segmentation, corrosion, and thinning. After converting these stripes into single-pixel lines, Figure 7b can be obtained. If the average horizontal spacing between these lines is denoted as *x* and the vertical spacing as *y*, the width *W* of moiré fringe can be calculated by
(8)$$W=\frac{x\cdot y}{\sqrt{x^{2}{+y}^{2}}}.$$

Substituting the width value *W* of the moiré fringe into (6) can yield the corresponding focal length value *f*_2_. Here, considering CCD pixels are discontinuous, some random errors may be introduced.

### 3.2. Experimental Results

The error source of this setup mainly comes from two parts, of which one is the CCD fringe fitting error, and the other is the beam drift caused by vibration and airstream in the optical system. Table 2 shows the corresponding focal length measurement results for the grating angle in the range of 2°–4°. The RMSEs of repeated measurements at 2°, 3°, and 4° are 2.5 mm, 3.9 mm, and 6.1 mm, respectively, and the corresponding permillage values are 1.9‰, 3.0‰, and 4.7‰, respectively. On the whole, the method has a higher measurement accuracy, and the measurement random error increases slightly as the grating angle increases. The test results demonstrate that it is feasible to use this setup for the focal length measurement.

### 3.3. Comparison and Discussion

At present, different schemes usually have different application scenarios. Table 3 focuses on the comparison of measurement accuracy and characteristics of various schemes. In terms of the accuracy indicator, the two schemes of Fizeau interferometer and the laser differential confocal method have extremely high measurement accuracy, but they are very sensitive to the working environment. Hence, there are few practical applications for these two schemes. At the same time, the laser differential confocal method is mainly used in some large optical systems, such as space optical systems, high-energy laser systems, and laser fusion programs [8,11]. The conjugate method is currently the most widely used solution owing to its simple design and easy implementation, and its disadvantage lies in the unstable measurement accuracy. Talbot interferometry is a relatively mature program, but it requires special design for different applications, and thus lacks flexibility. By contrast, our scheme ensures a higher measurement accuracy and requires no special working environment, so it has significant advantages in batch testing.

## 4. Conclusions

This paper focuses on the scheme of measuring the focal length of a collimating optical system with two gratings. According to the conjugate theory of objects and images in a symmetrical optical system, the relationship model between the moiré fringe and the focal length of the optical system is proposed, and theoretical simulation and experimental verification are carried out. The simulation results show that a larger difference between the focal lengths of two collimators can result in larger variation of the moiré fringe width. Moiré fringes have a good resolution ability for focal length differences within 5‰, especially at small angles below 4°. At the same time, the width difference at the symmetrical position of the focal length is small, indicating the poor discriminating ability of the proposed method at the symmetrical position of the reference collimator.

According to the experimental results, the overall measurement error of the proposed scheme is better than 4.7‰, and the accuracy is slightly improved within a small angle range of 3°. Therefore, the combination of gratings with moiré fringes can effectively amplify the difference between the focal lengths of collimators, thereby greatly improving the measurement accuracy of the collimator. Moreover, the variation of the moiré fringe imaging law caused by the focal length of the collimator is comprehensively analyzed, and the influence of the focal length difference can provide reference for other target measurements.

## Figures and Tables

**Figure 1 sensors-20-02718-f001:**
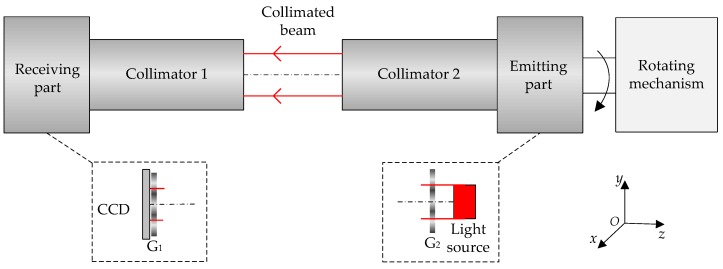
Schematic diagram of focal length measurement. CCD, charge coupled device.

**Figure 2 sensors-20-02718-f002:**
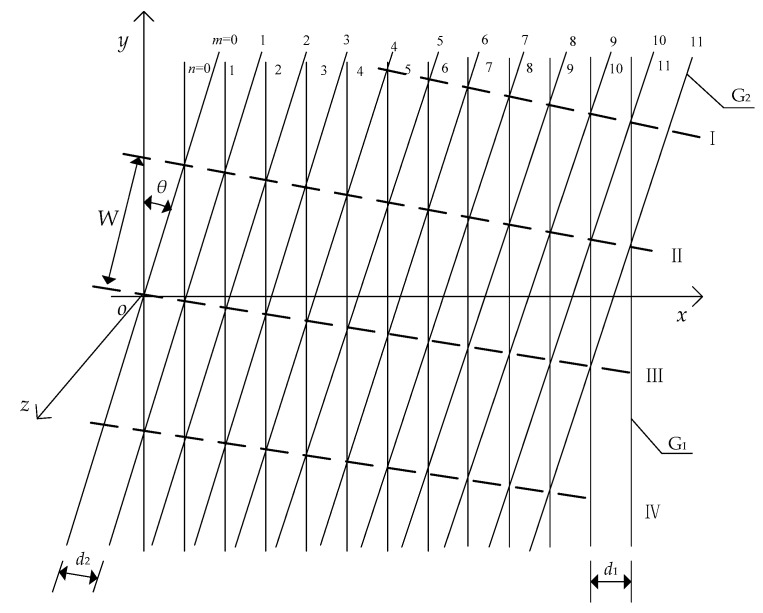
Moiré fringe coordinate system.

**Figure 3 sensors-20-02718-f003:**
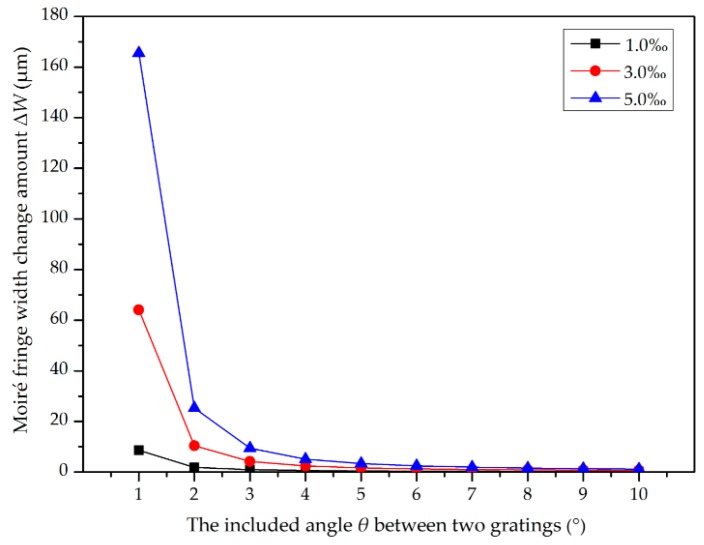
Curve of fringe width with focal length difference.

**Figure 4 sensors-20-02718-f004:**
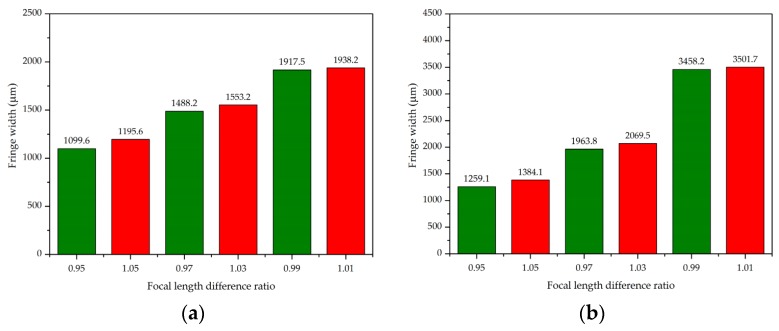
Variation of moiré fringe width at different focal length ratios of *f*_1_ and *f*_2_: (**a**) rotation angle of G_2_ is 1°, (**b**) rotation angle of G_2_ is 2°, where the green box represents *f*_2_ > *f*_1_ and the red represents *f*_2_ < *f*_1_.

**Figure 5 sensors-20-02718-f005:**
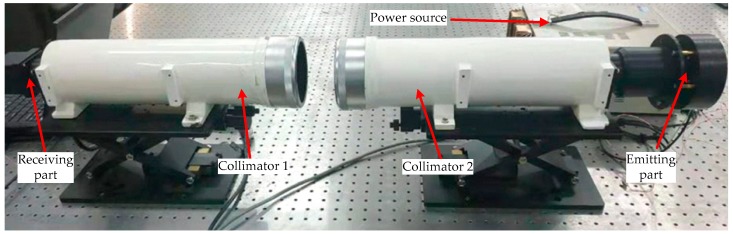
Layout of experimental setup.

**Figure 6 sensors-20-02718-f006:**
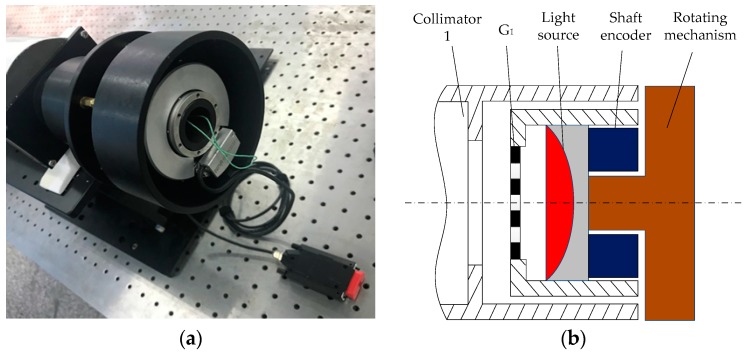
Structural principles of the emitting part: (**a**) physical installation, (**b**) principle structure.

**Figure 7 sensors-20-02718-f007:**
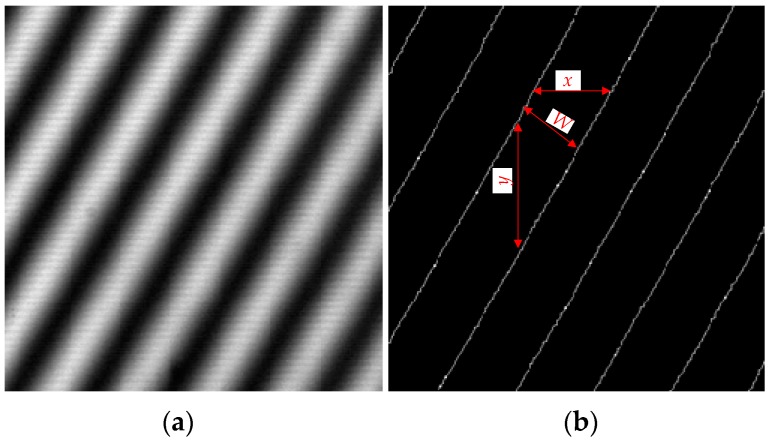
Moiré fringe image. (**a**) The original image; (**b**) extracted center lines.

**Table 1 sensors-20-02718-t001:** Main parameters of experimental setup. CCD, charge coupled device.

Property	Unit	Value
Wavelength of the red LED	nm	625
Focal length of collimator 1	mm	1325
Focal length of collimator 2	mm	1295
Grating pitch	μm	70
Grating diameter	mm	10
Pixel size of CCD	μm	4.8
Resolution of CCD	pixel	1999 × 2000
Leica theodolite	arcsec	±0.5
Shaft encoder MRP5080 accuracy	arcsec	0.1

**Table 2 sensors-20-02718-t002:** Measurement results (unit: mm).

Order	2°		3°		4°	
	Result	Error	Result	Error	Result	Error
1	1297.4	2.4	1298.3	3.3	1301.5	6.5
2	1296.9	1.9	1292.9	−2.1	1289.4	−5.6
3	1297.1	2.1	1298.7	3.7	1300.7	5.7
4	1293.7	−1.3	1300.5	5.5	1291.7	−3.3
5	1292.8	−2.2	1292.6	−2.4	1290.6	−4.4
6	1297.7	2.7	1299.8	4.8	1301.8	6.8
7	1298.9	3.9	1230.1	5.1	1303.2	8.2
8	1297.2	2.2	1299.5	4.5	1302.1	7.1
9	1298.1	3.1	1291.7	−3.3	1289.4	−5.6
10	1293.2	−1.8	1292.5	−2.5	1301.5	6.5

**Table 3 sensors-20-02718-t003:** Comparison of existing focal length measurement solutions.

Methods	Accuracy	Characteristics
Our scheme	1.9‰–4.7‰	Stable structural design, and good suitability for batch testing.
Conjugate	<5%	Wide application range, but unstable measurement accuracy.
Fizeau interferometer	<1‰	Good suitability for single lens, but high environmental requirements.
Talbot interferometry	0.5‰–2‰	Specially designed multi-parameter model for targeted purpose.
Laser differential confocal method	0.1‰–1‰	Good suitability for ultra-long focal length, but few applications.

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
