# Peer review of "Research on Focal Length Measurement Scheme of Self-Collimating Optical Instrument Based on Double Grating"

_sensors, 2020, doi:10.3390/s20092718_

Round 1

Reviewer 1 Report

The authors addressed all my previous comments very well and modified the paper in such a way that it can be recommended for publication.

Author Response

Thanks a lot for your review comments and all the help.

Reviewer 2 Report

The authors argue they propose a new method for measuring the focal length of a self-collimating optical instrument using gratings and moiré fringes.

Broad comments

The measurement of focal length is a subject studied over the past 30 years in works published in optics journals. Indeed, most of the reference used by the authors are from the area of Optics. In any case, I still found the work is little new and significant to be published in a journal such impacted as Sensors, and I cannot recommend it for its publication in this journal. Experimental results are limited just to one collimator and that is not robust enough to compare with other existing methods and draw the conclusions.

The simulation is performed with two collimators with the focal length difference of 1‰, 3‰, 5‰. However, the difference of the focal lengths used in the experiment is above 2%. The simulation and the experiment should match.

Specific comments

Line 147. “Therefore, the simulation results show that the focal length of collimator can be accurately measured by using the amplification characteristics of moiré fringes”. Avoid vague statements and define accuracy

Consider replacing figure 4 with a table

Line 124: Equation (3) is taken into (1) to obtain the moiré fringe width relationship formula with focal length factor. It is not clear how (3) is taken into (1) to get (4)

Line 175: Talbot instead “tablot”

Line 223: permillage instead of percentage

Author Response

Thanks a lot for your review comments and all the help. Please see the attachment for the revised manuscript.

Reviewer 3 Report

Review to manuscript #780960 entitled “Research on focal length measurement scheme of self-collimating optical instrument based on double grating” submitted to Sensors.

I had the opportunity of reviewing the previous version of the present manuscript and I have to admit that the authors have improved greatly the quality and expression of the paper.

On the other side, I still have some concerns that the authors should fix and explain before suggesting the manuscript for publication in sensors.

  • Firstly, despite the English expression and grammar have been deeply revised and corrected, I still have found some incongruent sentences and typos. Please, revise carefully again the manuscript to find and correct them. My mother language is not English either and I may understand the added difficulty but, this is a prestigious journal and manuscripts must be grammatically correct.

  • Secondly, I cannot see the difference between W (equation 1) and W’ (equation 4). They have to be different since you subtract them (equation 6) but I do not see the meaning of each one. This is an important point of the theoretical approach and I think it should be clearer.

I have tried to find out the meaning but I am not sure about it. Is W calculated when both focal lengths are equal and W’ when they are different?

I think the key point is in how you introduce equation 3 into equation 1 and define W’. It is also not clear since you define sigma_d and after that disappears in equation 4.

In addition, you define again W from the acquired images in equation 7, and say that you substitute it into equation 5, that has W’ instead W.

Clarify all this point, please.

  • Subtitle 2.3 should be 2.2.

  • Congratulation for the Chinese national invention patent but, is it necessary to mention it as a single section?

Based on my comments, I will recommend the manuscript for publication after having the opportunity of review it again and corroborate the answering of all my queries.

Author Response

(The authors gave the same response as above.)

Round 2

Reviewer 2 Report

The authors addressed all my comments and modified the paper in such a way that it can be recommended for publication.

Reviewer 3 Report

After three reviews, the authors have answered all my questions and solved all my concerns. In addition, English expression has been also greately improved. So, I recommend the publication of the manuscript in the present form.

This manuscript is a resubmission of an earlier submission. The following is a list of the peer review reports and author responses from that submission.

Round 1

Reviewer 1 Report

The paper is well written and organized. I would recommend to introduce or reference Eq. (3) better.

Reviewer 2 Report

The authors argue they propose a new method for measuring the focal length of a self-collimating optical instrument using gratings and moiré fringes. The measurement of focal length is a subject studied in depth over the past 30 years in works published in optics journals. Therefore, the work is little new and significant, and I cannot recommend it for its publication in this journal. Results are quite limited and are not robust enough to draw the conclusions. Moreover, I feel a thorough literature review is lacking. However, I would suggest authors to the authors that they send the article to a journal in the area of Optics after solving these comments.

Reviewer 3 Report

The paper presents new idea of using moiré effect to asses difference of the two collimators focal length. The method is well described and experimentally validated. The paper is ready for publication, but I have still few minor questions.

a) The chapter 3 mentions using theodolite for optical axes alignment. There is no information about the instrument, its uncertainly or residual misalignment error.

b) Authors mentioned that: "the method has poor discriminating ability at the symmetrical position of the reference".  How did you distinguished that the measured collimator has shorter focal length then reference?

c) What is uncertainty of determination of moiré fringes width and how it affects uncertainty of the result?

      Even with those questions I assume the paper acceptable for publishing.

Reviewer 4 Report

Review to manuscript 628272 entitled “Research on focal length measurement method of self-collimating optical instrument based on double grating” submitted to Sensors.

General remarks:

In this work, the authors probe the usage of Moiré fringes to measure the focal length of a lens or group of lenses (collimating optical instrument). In the first sentence of the abstract, the authors claim that they are proposing a new method and I sincerely do not agree.

A rapid search gives a lot of references related to focal length and moiré effect. Besides references [14] and [15], the authors have forgotten to include some important references that evidence the usage of Moiré fringes to obtain the focal length of a lens such as:

Glatt and O. Kafri, “Determination of the focal length of nonparaxial lenses by more deflectometry”, App. Opt. 26 (13), 1987.

Ching-Huang Lin et al. “Focal length measurement by the analysis of moiré fringes using the wavelet transformation”, Journal of the Chinese institute of engineers, 28(1), 2005.

De Angelis et al. “Analysis of moiré fringes for measuring the focal length of lenses”, Opt. Laser Eng. 30(3), 1998.

On the other hand, the authors do not explain how to measure the period of the moiré fringes and I think it is very important for the accuracy of the method. Refer, for example, to

F.J. Torcal-Milla and L.M. Sanchez-Brea, “Near-field diffraction-based focal length determination technique”, Opt. Laser Eng. 92, 2017

Anyway, the authors should compare their results with other methods and give clearly the advantages of this method against others that use moiré fringes or other phenomenology.

Other remarks:

In my opinion, the manuscript is hard to be read. Several sentences are grammatically incorrect. I recommend a deep review of the manuscript to improve its English expression. It is not clear what equation (4) represents. Does it correspond to a rotated angle of the second grating? How do you obtain it? Explain, please. What is a negative lens collimator? What is the purpose of such an optical element? The experimental section is poorly explained. I think it should be the more important part of the manuscript. The authors do not explain how is the light source, where is it placed, and so on? It should be placed at back focal length of the first collimator but, is it important? LED are not point sources, so some effect could be also noticed. What do you think about that? At the end of page 3, I think it should be “…the moiré period w” instead of “… the grating period d”. In page 6, after table 1, it should be “As shown in Figure 5…” instead “ Figure 7”.

My impression is that the authors have employed a lot of time in developing a method that was already done, at least, partially. I am sorry of not giving you a kinder response but, in my opinion, the concept is not new and therefore, I recommend rejection of the manuscript.